# Seq2Neo: A Comprehensive Pipeline for Cancer Neoantigen Immunogenicity Prediction

**DOI:** 10.3390/ijms231911624

**Published:** 2022-10-01

**Authors:** Kaixuan Diao, Jing Chen, Tao Wu, Xuan Wang, Guangshuai Wang, Xiaoqin Sun, Xiangyu Zhao, Chenxu Wu, Jinyu Wang, Huizi Yao, Casimiro Gerarduzzi, Xue-Song Liu

**Affiliations:** 1School of Life Science and Technology, ShanghaiTech University, Shanghai 201203, China; 2Shanghai Institute of Biochemistry and Cell Biology, Chinese Academy of Sciences, Shanghai 200031, China; 3University of Chinese Academy of Sciences, Beijing 100049, China; 4Département de Médecine, Faculté de Médecine, Université de Montréal, Montréal, QC H4T 1G2, Canada

**Keywords:** immunogenicity, immunotherapy, bioinformatics pipeline, deep learning

## Abstract

Neoantigens derived from somatic DNA alterations are ideal cancer-specific targets. In recent years, the combination therapy of PD-1/PD-L1 blockers and neoantigen vaccines has shown clinical efficacy in original PD-1/PD-L1 blocker non-responders. However, not all somatic DNA mutations result in immunogenicity among cancer cells and efficient tools to predict the immunogenicity of neoepitopes are still urgently needed. Here, we present the Seq2Neo pipeline, which provides a one-stop solution for neoepitope feature prediction using raw sequencing data. Neoantigens derived from different types of genome DNA alterations, including point mutations, insertion deletions and gene fusions, are all supported. Importantly, a convolutional neural network (CNN)-based model was trained to predict the immunogenicity of neoepitopes and this model showed an improved performance compared to the currently available tools in immunogenicity prediction using independent datasets. We anticipate that the Seq2Neo pipeline could become a useful tool in the prediction of neoantigen immunogenicity and cancer immunotherapy. Seq2Neo is open-source software under an academic free license (AFL) v3.0 and is freely available at Github.

## 1. Introduction

In recent years, PD-1/PD-L1 blocker immunotherapy has transformed the treatment of cancer. PD-1 is a protein found on T cells that helps to keep the immune systems in check. The combination of PD-1 and PD-L1 helps to stop T cells killing other cells, including cancer cells, which can result in immune evasion [1,2,3,4]. Previous studies have reported that only a small proportion of patients present lasting clinical responses while most patients only present transient responses or no response at all [5,6]. The combination of PD-1 blockers and other forms of immunotherapy, such as neoantigen vaccines, has demonstrated favorable development prospects [7,8].

Neoantigens derived from somatic DNA alterations are ideal cancer-specific targets. Neoantigen vaccines have demonstrated therapeutic effects in terms of enhancing immunotherapy efficacy [9]. It has also been reported that the combination of PD-1 antibodies and neoantigen vaccines is safe and effective in the treatment of cancer patients [8]. In addition, TCR-T-targeting neoantigens have shown dramatic effects in clinical practice [10]. However, the success of these neoantigen-related therapies relies on efficient neoantigen prediction tools.

A plethora of peptide–HLA binding prediction algorithms have been developed to predict which peptides would bind to specific cognate HLA alleles [11,12,13,14]. However, HLA–peptide binding affinity alone is not sufficient for predicting the immunogenicity of peptides. In addition, the currently available neoantigen prediction tools only provide limited neoantigen features or focus on specific genome alterations, such as point mutations. Methods for the accurate prediction of the immunogenicity of neoantigens based on raw sequence data are still urgently needed. Here, we present an open-source pipeline tool, Seq2Neo, which could provide a one-stop service for raw data preprocessing, HLA typing, mutation labeling and neoantigen prediction as it can support neoantigens derived from point mutations, insertion and deletions (INDELs) and gene fusions and predict various neoantigen features for each candidate peptide, including HLA binding affinity, the transport efficiency of transporters associated with antigen processing (TAP) and gene expression. Importantly, a convolutional neural network (CNN)-based immunogenicity prediction model was also constructed and this model showed an improved performance compared to other known methods.

## 2. Results and Discussion

### 2.1. Neoantigen Feature Prediction

In our study, Seq2Neo used a command line-based interface, which allowed users to perform workflows automatically. Seq2Neo used publicly available tools for mutation labeling, HLA typing and HLA affinity binding prediction. Then, a CNN-based model was constructed with features that were generated using Seq2Neo to predict the immunogenicity of peptides and directly stimulate CD8+ T cell response. Finally, Seq2Neo outputted various peptide features, including immunogenicity score, peptide–HLA binding affinity, TAP transport efficiency and gene expression (Figure 1 and Appendix A). The Seq2Neo model (Figure 1) began by importing raw sequencing data in FASTQ, SAM or BAM format and then utilized the user input to select the corresponding workflow to run. Point mutation and INDEL detection was performed using Mutect2 [15] and gene fusion detection was performed using STAR-Fusion [16]. Subsequently, somatic variant data were generated in VCF format. MHC genotyping was performed using HLA-HD [17]. Before the neoantigen prediction, sample somatic variants were annotated using ANNOVAR [18] or Agfusion [19] to obtain potential mutant peptides.

### 2.2. Selection of the Best HLA-I Binding Affinity Prediction Algorithms

We used 23319 peptides (14677 positives, with IC50 = 500 nm as the threshold) from the Immune Epitope Database (IEDB) to evaluate the performance of selected peptide–HLA I binding affinity prediction algorithms, including NetMHCpan [20], MHCflurry [21], PickPocket [22] and NetMHCcon [23]. The NetMHCpan BA model obtained the highest accuracy score (0.75) and the highest precision score (0.96) (Figure 2A,B), so this algorithm was selected for peptide–HLA binding prediction in Seq2Neo. In addition to peptide–HLA binding affinity, Seq2Neo used TPMCalculator [24] to detect gene expression and NetCTLpan [25] to obtain TAP transport efficiency.

### 2.3. Data Used for Seq2Neo-CNN Model Training

The fundamental feature of neoepitopes is their ability to stimulate cytolytic T cell responses, but this immunogenicity information cannot be predicted using most of the current neoantigen prediction tools. For immunogenicity prediction, we searched the IEDB database for experimental evidence that supported the immunogenicity of peptides and acquired 75496 experimentally evaluated immunogenicity assays (Figure 3). After applying our filter criteria (Section 3), 8975 data points (5342 negative peptides) were retained in the final dataset. We chose an independent dataset for model validation, which included 599 experimentally tested tumor-specific neoantigens from the Tumor Neoantigen Selection Alliance (TESLA) after deduplication and length restriction to 8–11 [26].

### 2.4. Features Associated with Peptide Immunogenicity

In order to find beneficial features for immunogenicity prediction, we compared the features of immunogenic and non-immunogenic peptides. The features of HLA-binding affinity, TAP transport efficiency and proteasomal C terminal cleavage were considered. The differences in HLA-binding affinity and TAP transport efficiency between the immunogenic and non-immunogenic peptides were significant but those in proteasomal C terminal cleavage were not (Figure 4A–C). HLA-binding affinity and TAP transport efficiency were not correlated (R = 0.02 and P = 0.055; Figure 4D); therefore, we incorporated these two features into our Seq2Neo-CNN model.

### 2.5. Seq2Neo-CNN Model for Immunogenicity Prediction

We built a CNN-based model, named Seq2Neo-CNN, to predict peptide immunogenicity (Figure 5A). The performance of the trained CNN model was compared to that of other machine learning models (ExtraTree, random forest, logistic regression, SVM and XGBoost), which were trained with the prediction accuracy, recall and precision data that were collected in this study (Figure 5B) and also data from the independent TESLA dataset (Appendix A). The Seq2Neo-CNN model showed the highest performance compared to the other machine learning models. The performance of Seq2Neo-CNN was also compared to other available neoepitope immunogenicity prediction tools using the independent TESLA dataset, including the DeepHLApan [27], IEDB [28] and DeepImmuno-CNN [29] models. Seq2Neo-CNN also showed the highest performance compared to these selected known methods (Figure 5C). The details of the Seq2Neo-CNN model construction and training are described in Section 3.

### 2.6. Seq2Neo Validation

In recent years, several tools for predicting neoantigens have been reported. Some representative tools are shown in Table 1 [12,14,30,31,32,33]. Two of the pipelines (TSNAD2 and Neopepsee) contain immunogenicity prediction functions. However, the other tools call their immunogenicity prediction modules DeepHLApan and IEDB, which proved to be less accurate than Seq2Neo-CNN (Figure 5C). Compared to the other pipelines, ease of use was also an advantage of Seq2Neo, since the Seq2Neo model provides a one-stop solution for neoantigen prediction using raw sequencing data. To demonstrate the performance of Seq2Neo, we applied Seq2Neo to samples from five cancer patients with experimentally validated neoantigenic mutations [34,35,36,37]. Those cancer samples contained WES, RNA-seq data and 16 experimentally validated neoantigenic DNA sites (Appendix A). After applying the selection criteria (TAP > 0, IC50 ≤ 500, TPM > 0 and immunogenicity > 0.5), Seq2Neo identified 10 out of the 16 validated neoantigenic sites. The ranking of the candidate neoantigens is shown in Appendix A. We selected three pipelines with detailed documentation, namely pVACseq, TSNAD 2.0 and NeoPredPipe to compare to Seq2Neo. Then, we compared the prediction results of Seq2Neo to those of the other three pipelines. The ranking of the most validated neoantigenic sites in Seq2Neo was lower than that in the other three pipelines, which meant that Seq2Neo demonstrated an improved performance in terms of identifying the real immunogenic neoantigens (Appendix A).

### 2.7. Seq2Neo Implementation

The Seq2Neo pipeline was developed in Python 3.7.12 following a clean, modular and robust design, in accordance with best practice coding standards. The instructions for installing and running Seq2Neo are presented in a public GitHub repository (https://github.com/XSLiuLab/Seq2Neo accessed on 28 June 2022). This model was designed to run as a command line-based program with a user-friendly interface, thereby allowing non-expert users to become familiarized with its functions quickly. To facilitate the installation of Seq2Neo, Docker containers and Conda packages are provided (Docker: https://hub.docker.com/r/liuxslab/seq2neo accessed on 28 June 2022; Conda: https://anaconda.org/liuxslab/seq2neo accessed on 28 June 2022).

## 3. Materials and Methods

### 3.1. Data Preprocessing

The Seq2Neo model began by importing data in FASTQ, SAM and BAM format and then utilized the user input to select the corresponding workflow to run. The FASTQ files were processed for quality control and any adapter sequences at the end of the reads were removed using Fastp [38]. The raw sequence data were aligned to the reference genome (hg38) using the Burrows–Wheeler alignment tool [39]. When the input format was SAM or BAM, GATK best practice was performed first during the data preprocessing [40]. The SAM files were sorted and read group tags were added using Samtools [40]. After being sorting into coordinate order, the BAM files were processed using PICARD MarkDuplicates and the local realignment and quality score recalibration were conducted using the Genome Analysis Toolkit [41].

### 3.2. Somatic Mutation Detection

Generated or user-inputted co-cleaned BAM files were used for point mutation and insertion and deletion (INDEL) detection using Mutect2 [15] and gene fusions were detected using STAR-Fusion [16]. Then, somatic variant data were generated in VCF format. Additionally, parallel computation was enabled, which significantly reduced the computation time.

### 3.3. HLA Genotyping

Human leukocyte antigen (HLA) genes play a critical role in antigen presentation and immune signaling. Here, HLA-HD [17] was adapted for HLA genotyping using DNA-seq data and outputted personal HLA types for each patient, including class I and II HLAs.

### 3.4. Gene Expression Detection

The expression and presentation of tumor antigen-presenting cells on the surface are the prerequisites for neoantigens to be recognized by T cells. Seq2Neo supported the annotation of the expression of neoantigen candidates using TPMCalculator [24].

### 3.5. Neoepitope Features

In addition to peptide–HLA binding affinity, other features, including TAP transport efficiency, gene expression and immunogenicity score, could also be predicted using Seq2Neo. These neoepitope features could facilitate the filtering of candidate peptides for vaccine or immunotherapy target selection.

### 3.6. Immunogenicity Prediction (Seq2Neo-CNN Model)

As the core of the Seq2Neo pipeline, Seq2Neo-CNN could predict the immunogenicity of selected peptides. Below, we provide a detailed description of the generation of the Seq2Neo-CNN model.

#### 3.6.1. Dataset Selection

We collected data from the IEDB database for the initial model training and validation (3 August 2021 version) using the following IEDB searching conditions: epitope (linear sequence), assay (positive/negative), T cell assay, MHC restriction (MHC class I), host (Human) and disease (any). In all, we found 75,496 relevant experiments. Although there are different ways to detect the immunogenicity of peptides, some experiments did not detect direct contact with T cells that induced immune responses, so we only selected data that were validated by ELISPOT, 51 Chromium, ICS, Multimer/Tetramer and ELISA. Then, we deleted any instances that did not have four-digit MHC alleles or were repeated. We also limited the length of peptides to 8–11 mer and removed negative peptides that had missing experimental information or less than four test subjects. Finally, we obtained 8975 peptides that met the requirements for the final dataset, among which 3633 were positive reactive instances and the remaining 5342 were negative. We selected an independent dataset for further evaluation, which included 599 experimentally tested tumor-specific neoantigens from the Tumor Neoantigen Selection Alliance (TESLA) after selecting only 8–11 mer peptides and removing duplicates [26].

#### 3.6.2. Allele Representation

In order to input the MHC class I alleles into the neural network in numerical matrix form, we used pseudo-sequences to represent them. The pseudo-sequences were constructed by Nielsen et al. [42] and consisted of amino acid residues that were in contact with the peptides. The selected positions were 79, 24, 45, 59, 62, 63, 66, 67, 69, 70, 73, 74, 76, 77, 80, 81, 84, 95, 97, 99, 114, 116, 118, 143, 147, 150, 152, 156, 158, 159, 163, 167 and 171. We used the following strategy to encode the MHC pseudo-sequences.

#### 3.6.3. Encoding Strategy

We used a one-hot encoding scheme to represent each HLA allele and peptide sequence in numerical matrix form, which were used as the inputs for the following algorithms. The one-hot encoding scheme was realized by assigning a unique integer to each letter in the 21-digit amino acid alphabet that contained padding characters as the index of that letter in the amino acid alphabet. Taking the letter “A” as an example, we obtained the alphabet “ACDEFGHIKLMNPQRSTVWYX” (the unknown amino acid was set to “X”) and the corresponding index of alanine “A” was 0. Then, the values of the other amino acids were set to 0, but the value of “A” was set to 1. Finally, we obtained the one-hot vector of [1, 0, 0, 0, 0, 0, 0, 0, 0, 0, 0, 0, 0, 0, 0, 0, 0, 0, 0, 0, 0]. For each peptide, the unique one-hot vectors of each amino acid in the amino acid sequence were vertically combined to form a numerical matrix to complete vectorization.

#### 3.6.4. Feature Normalization

Binding affinity and TAP transport efficiency were predicted for all peptide–HLAs using the method that was described previously and then normalized using the maximum and minimum values simultaneously. The basic mathematical form was represented as:$$y=\frac{\left(x-x_{min}\right)}{\left(x_{max}-x_{min}\right)}$$

#### 3.6.5. Prediction Model

We used a CNN (convolutional neural network) to predict the immunogenicity of mutant peptides. The proportions of the training set, testing set and validation set were 70%, 20% and 10%, respectively. The peptides and MHCs were processed by two consecutive convolutional layers, followed by three dense layers to execute the affine transformation and then flattened vectors with dimensions of 256 were obtained. NetMHCpan and NetCTLpan were used to calculate the binding affinity (IC50) and TAP transport efficiency of the peptides and those features were used to train the natural network. To incorporate the IC50 and TAP transport efficiency features into our CNN, two dense layers were included. Finally, Seq2Neo outputted the immunogenicity prediction. We used the ReLu function as the activation function. Some hyperparameters were set during the optimization process before the training started: batch size was set to 64, training loss with patience was set to 15, validation loss with patience was set to 20, epochs were set to 200 and the Adam learning rate was set to 0.001. Two early stopping strategies were adopted to ensure that the acquired model was the best possible version. In addition, we adopted batch normalization and dropout strategies to accelerate the model convergence speed and enhance its generalization ability. Since the number of negative reactive instances was significantly higher than that of positive reactive instances, the weight was set according to the proportions of negative and positive instances to eliminate this imbalance. The weight operation was mathematically represented as:$$w=\frac{1}{S}\times \frac{T}{2}$$
where *w* is the negative or positive class weight, *S* is the number of corresponding reactive instances and *T* is the total number of training instances.

### 3.7. Other Machine Learning-Based Immunogenicity Prediction Models

In order to select the best model to predict immunogenicity, we compared the Seq2Neo-CNN model to five other machine learning algorithms (logistic regression, SVM, XGBoost, random forest and ExtraTree) after optimizing the parameters for each method. We used accuracy as the evaluation criterion to tune the best parameters for each model. The best parameters for logistic regression were acquired through 10-fold cross-validation (penalty = l2 and C = 2.21). Similar to logistic regression, kernel = rbf, gamma = 0.1 and C = 10 were the best parameters for SVM., whereas max_depth = 10, min_child_weight = 1.0, gamma = 1.625, subsample = 1.0 and colsample_bytree = 1.0 were the best parameters for XGBoost, n_estimators = 200 and min_samples_leaf = 2 were the best parameters for random forest and n_estimators = 1000 and min_samples_leaf = 2 were the best parameters for ExtraTree. Then, the optimized models were compared to the Seq2Neo-CNN model using the testing set and the TESLA dataset.

### 3.8. Seq2Neo Implementation in Cancer Patient Samples

To test the performance of the Seq2Neo pipeline, WES (normal/tumor exome) and RNA-seq (tumor transcriptome) data from five patients with different solid tumors were downloaded from the NCBI SRA database (bioproject IDs: PRJNA298310, PRJNA298330 and PRJNA298376) [34,35,36]. Each sample had 2–4 experimentally verified neoantigens derived from point mutations that could induce T cell responses. Here, we used Seq2Neo to predict these neoantigens to verify the performance of Seq2Neo. Then, we compared the rank percentage of Seq2Neo to that of pVACseq using default parameters.

## 4. Conclusions

As a supplement to PD-1 immunotherapy, neoantigens are ideal cancer-specific targets for precision vaccine design or TCR-T therapy and act as key factors in cancer immunoediting [43]. However, current neoantigen prediction is cumbersome and lacks a comprehensive one-step tool. Furthermore, most neoantigen prediction tools only focus on the binding between peptides and HLA I and accurate tools for directly predicting the immunogenicity of neoepitopes are still lacking. Seq2Neo is a user-friendly and robust tool that could provide a one-stop solution for neoantigen prediction using raw sequencing data. Importantly, various features of neoantigens can be predicted using Seq2Neo, including the immunogenicity capability of neoepitopes.

## Figures and Tables

**Figure 1 ijms-23-11624-f001:**
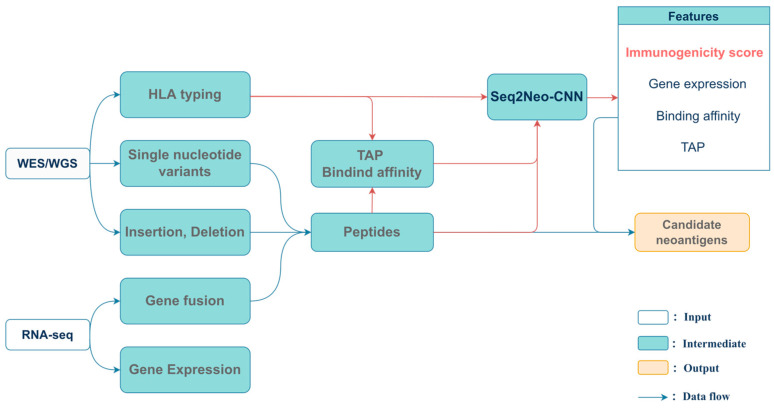
An overview of Seq2Neo. The input of Seq2Neo includes raw WGS, WES, RNA-seq or peptide information. Seq2Neo predicts various peptide features, including CNN-based immunogenicity score, peptide–HLA binding affinity, TAP transport efficiency and gene expression. Then, Seq2Neo uses those features to rank candidate neoantigens.

**Figure 2 ijms-23-11624-f002:**
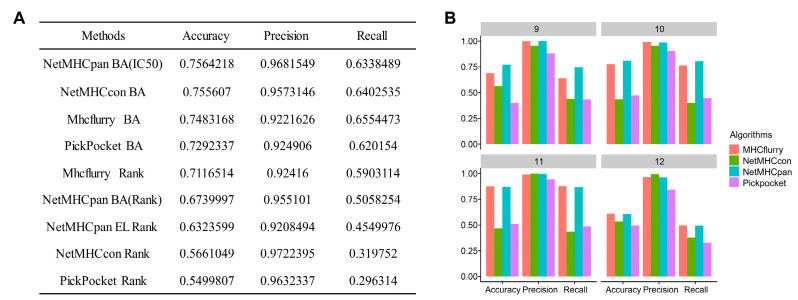
A benchmark analysis of different HLA-I binding affinity prediction algorithms: (**A**) a comparison of the performance of different algorithms in terms of prediction accuracy, precision and recall (the dataset was downloaded from IEDB and the thresholds of an IC50 value less than 500 nM and a rank percentile less than 1% were used to determine positive peptides); (**B**) a comparison of the different algorithms on different lengths of peptides.

**Figure 3 ijms-23-11624-f003:**
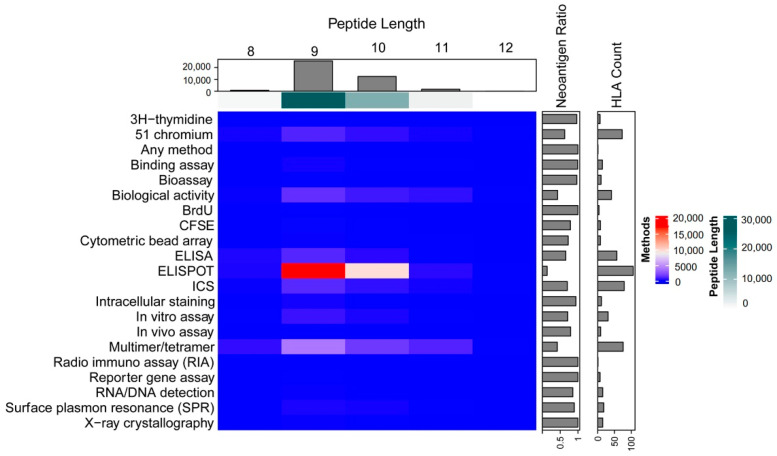
An overview of the data that were used for the Seq2Neo model training, including basic information about the IEDB dataset. We restricted the dataset to peptides with metadata that matched the following keywords: (1) linear epitopes, (2) specific T cell assays, (3) intact MHC I class, (4) originated from humans, (5) any diseases and (6) intact test information for negative peptides. CFSE, carboxyfluorescein succinimidyl amino ester; ELISA, enzyme-linked immunosorbent assay; ELISPOT, enzyme-linked immunosorbent spot; ICS, intracellular cytokine staining.

**Figure 4 ijms-23-11624-f004:**
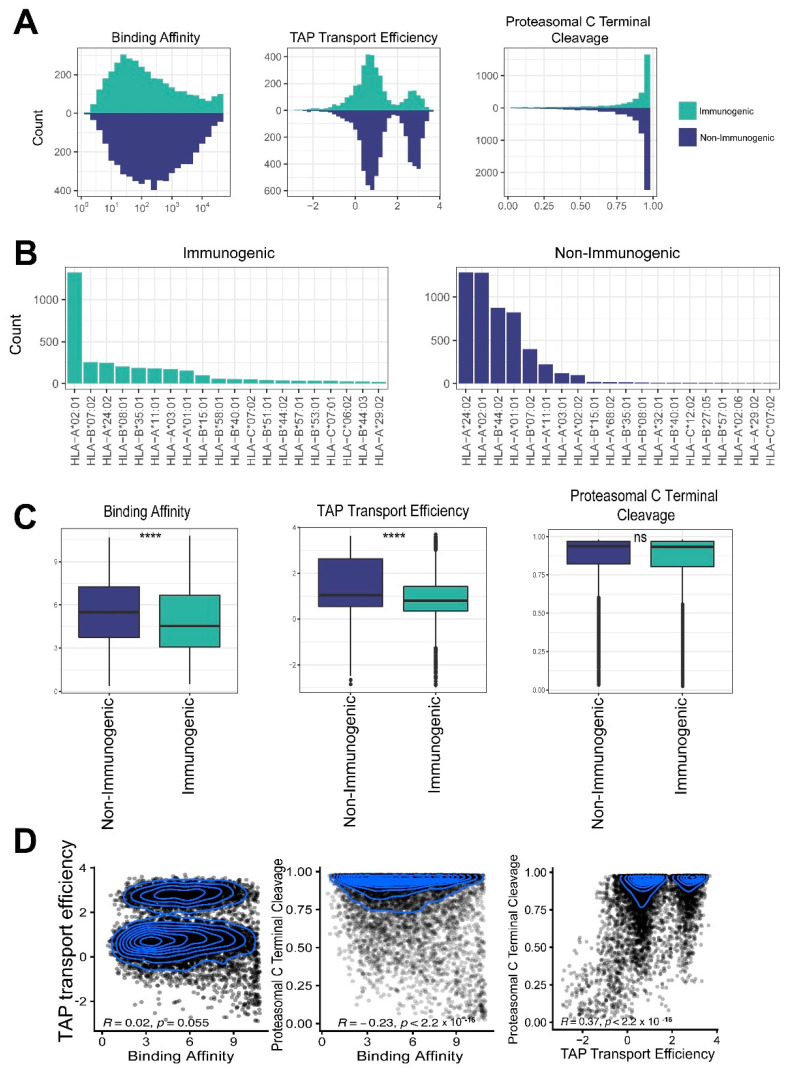
An exploration of the feature differences between immunogenic and non-immunogenic peptides: (**A**) a distribution comparison of HLA I binding affinity, TAP transport efficiency and proteasomal C terminal cleavage between immunogenic and non-immunogenic peptides; (**B**) a distribution comparison of the binding of the 20 most frequent HLA-I alleles to the immunogenic (left) and non-immunogenic mutated peptides (right); (**C**) a comparison of binding affinity, TAP transport efficiency and proteasomal C terminal cleavage between immunogenic and non-immunogenic mutated peptides (****, *p* < 10^−4^; ns, not significant); (**D**) pairwise correlations between the three neoepitope features (peptide–HLA binding affinity, TAP transport efficiency and proteasomal C terminal cleavage), showing the Pearson correlation coefficients R and *p* values.

**Figure 5 ijms-23-11624-f005:**
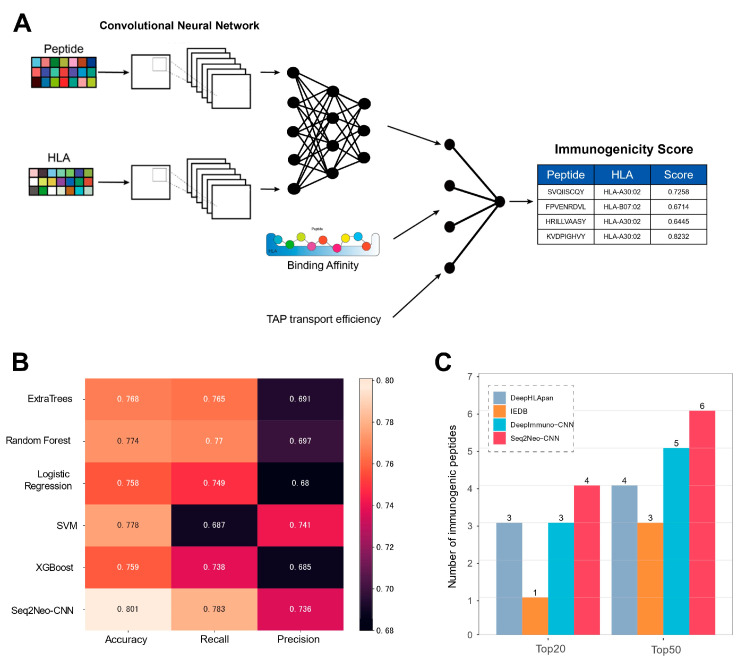
Our convolutional neural network-based model (Seq2Neo-CNN) for peptide immunogenicity prediction: (**A**) a schematic diagram of the Seq2Neo-CNN architecture (in this model, each peptide–MHC pair is subjected to two consecutive convolutional layers, followed by three fully connected dense layers and is then input into two fully connected dense layers, together with TAP transport efficiency and binding affinity information, to output a predicted immunogenicity value); (**B**) a comparison between the Seq2Neo-CNN model and other machine learning algorithms (to select the best predictive model, we constructed five traditional machine learning classifiers (ExtraTree, random forest, logistic regression, SVM and XGBoost) and the accuracy, recall and precision of each method are shown); (**C**) a comparison of the performance of the different models when predicting immunogenic peptides, based on the number of true positive peptides that overlapped with the top 20 or 50 predictions of each algorithm (the Seq2Neo-CNN model outperformed the existing immunogenicity prediction methods using the independent TESLA dataset).

**Table 1 ijms-23-11624-t001:** Representative tools for predicting neoantigens that have been published in recent years. The neoantigen type, input data, neoantigen class, HLA typing, immunogenicity score, TAP score and programming language that were used are presented.

Method	Neoantigen Types	Input Data	Neoantigen Class	HLA Typing	Immunogenicity Score	TAP Score	Language	Publish Year
Seq2Neo	SNVs, indels, gene fusions	WES/WGS, RNA-seq	Class I	Yes	Yes	Yes	Python	This study
pVACseq	SNVs, indels, gene fusions	VCF	Class I and II	No	No	No	Python	2019
TSNAD 2	SNVs, indels, gene fusions	WES/WGS, RNA-seq	Class I	Yes	Yes	No	Python	2021
NeoPredPipe	SNVs, indels	VCF, HLA types	Class I and II	No	No	No	Python	2019
Neopepsee	SNVs	VCF, RNA-seq, HLA types	Class I	Yes	Yes	No	Java	2018
nextNEOpi	SNVs, indels, gene fusions	WES/WGS, RNA-seq	Class I and II	Yes	No	No	Nextflow	2021
ProTECT	SNVs	WES/WGS, RNA-seq	Class I and II	Yes	No	No	Python	2020

## Data Availability

The HLA-I binding affinity and immunogenicity data were downloaded from the IEDB data portal (https://www.iedb.org/, accessed on 3 August 2021). The WES data and mRNA expression data from the five cancer samples were downloaded from the SRA database of NCBI (https://www.ncbi.nlm.nih.gov/sra, accessed on 20 May 2022).

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
