# Peer review of "Seq2Neo: A Comprehensive Pipeline for Cancer Neoantigen Immunogenicity Prediction"

_ijms, 2022, doi:10.3390/ijms231911624_

Round 1

Reviewer 1 Report

The authors present the Seq2Neo pipeline for neoepitope feature prediction from raw sequencing data. The paper is well-written and scientifically sound, but I have some remarks.

Other tools with similar goals and features are mentioned in the introduction and results sections, and the Seq2Neo-CNN model is compared with other models, but how does the Seq2Neo tool compare to the other tools (NeoFox, NextNEOpi), in terms of usability, features, performance, programming language, etc.? A table would be useful.

What is the reasoning behind the inclusion of supplementary information? In my opinion, the performance of the Seq2Neo-CNN model (figure 2) is quite important, so this should be included in the paper itself.

Abstract:

- "combination therapy ... show" -> "combination therapy ... shows"

Section 3.6.5:

- "Seq2Neo output" -> "Seq2Neo outputs"

Author Response

Point 1: Other tools with similar goals and features are mentioned in the introduction and results sections, and the Seq2Neo-CNN model is compared with other models, but how does the Seq2Neo tool compare to the other tools (NeoFox, NextNEOpi), in terms of usability, features, performance, programming language, etc.? A table would be useful.

Response: Thanks for the point. We have added a new table to compare some representative pipelines (New Table 1), and the performance of Seq2Neo has been compared with pVACseq, TSNAD v2.0 and NeoPredPipe, which are shown in Figure S3C.

 Point 2: What is the reasoning behind the inclusion of supplementary information? In my opinion, the performance of the Seq2Neo-CNN model (figure 2) is quite important, so this should be included in the paper itself.

Response: Thanks for this point. As suggested, this supplementary information has been moved to the main paper (revised Figure 5B).

Ponit 3

Abstract:

- "combination therapy ... show" -> "combination therapy ... shows"

Section 3.6.5:

- "Seq2Neo output" -> "Seq2Neo outputs"

Response: Thanks for this point. As suggested, these mistakes have now been corrected in the revised manuscript.

Reviewer 2 Report

In this paper, the authors developed a new method named Seq2Neo for the prediction of neo-antigen immunogenicity. The method is well described, and the GitHub page is user-friendly. However, I have a few major concerns regarding the validation of their method.

Major comments

(1) The authors described that they validated their Seq2Neo method in the samples from five cancer patients with experimentally validated neoantigen mutations. However, the dataset for five patients is very small for validation. The authors should try to use more datasets in this regard.

(2)  The authors only compared the validation of their method with pVACseq. But, several other tools can also predict neoepitopes and neoantigen immunogenicity. Some of the important tools are ProTECT (https://github.com/BD2KGenomics/protect), Neopepsee (https://sourceforge.net/projects/neopepsee/), and NeoPredPipe (https://github.com/MathOnco/NeoPredPipe). All these tools should be considered to compare the prediction results of Seq2Neo. The authors should compare their method with these tools and highlight why their pipeline/method is more effective than these tools/pipelines. They should compare which features are not available in other tools and those are available in their tool. A table should be generated for such a comparison, which should be included in the main manuscript draft, and the comparison results should be provided in the supplementary files.

Author Response

Point 1: The authors described that they validated their Seq2Neo method in the samples from five cancer patients with experimentally validated neoantigen mutations. However, the dataset for five patients is very small for validation. The authors should try to use more datasets in this regard

Response: Thanks for this point. We agree with the reviewer that five patients are a pretty small number for validation. Actually, the performance of Seq2Neo has also been compared with other tools in the independent TESLA dataset, which contains 6 patients and 608 experimentally tested epitopes. To obtain additional datasets with experimentally confirmed neoantigen immunogenicity information is not practical at the current stage.

Point 2: The authors only compared the validation of their method with pVACseq. But several other tools can also predict neoepitopes and neoantigen immunogenicity. Some of the important tools are ProTECT, Neopepsee, and NeoPredPipe. All these tools should be considered to compare the prediction results of Seq2Neo. The authors should compare their method with these tools and highlight why their pipeline/method is more effective than these tools/pipelines. They should compare which features are not available in other tools and those are available in their tool. A table should be generated for such a comparison, which should be included in the main manuscript draft, and the comparison results should be provided in the supplementary files

Response: Thanks for this essential point. As suggested, we have added a new table to compare the features of several representative pipelines (New Table 1), and the performance of Seq2Neo has also been compared with pVACseq, TSNAD v2.0 and NeoPredPipe using cancer patients’ sequencing data (Figure S3C).

Round 2

Reviewer 2 Report

The authors now compared the validation of their Seq2Neo method with pVACseq, TSNAD 2.0, and NeoPredPipe. But, several similar tools mentioned in Table 1 are still not considered for validation. Why did the authors not consider ProTECT and Neopepsee for validation? The data for validation results should be provided in the supplementary table. Moreover, the authors did not properly discuss why their method is better than other similar tools/methods listed in Table 1 and why the users should use their tools instead of other similar tools. I would recommend the authors consider these points before the publication of this manuscript.

Author Response

Point 1: The authors now compared the validation of their Seq2Neo method with pVACseq, TSNAD 2.0, and NeoPredPipe. But, several similar tools mentioned in Table 1 are still not considered for validation. Why did the authors not consider ProTECT and Neopepsee for validation? The data for validation results should be provided in the supplementary table. Moreover, the authors did not properly discuss why their method is better than other similar tools/methods listed in Table 1 and why the users should use their tools instead of other similar tools. I would recommend the authors consider these points before the publication of this manuscript.

Response: Thanks for this critical point. ProTECT tool does not have an immunogenicity prediction module, and Neopepsee applies IEDB tool for immunogenicity prediction, which has already been demonstrated to be an inaccurate immunogenicity prediction tool. Thus as an example, only pVACseq, TSNAD 2.0 and NeoPredPipe have been applied in the comparison study (Figure S3C). The focus of our work is to establish an immunogenicity prediction model and integrate it into an easy-to-use pipeline, namely Seq2Neo. In table 1, except for TSNAD 2.0 and Neopepsee, other pipelines do not have the immunogenicity prediction function. TSNAD 2.0 and Neopepsee call DeepHLApan and IEDB, respectively, as their immunogenicity prediction modules, which have been proved to be less accurate than Seq2Neo-CNN in Figure 5C. Together, the two highlights of Seq2Neo compared to known tools are 1, the most accurate immunogenicity prediction; 2, providing an easy-to-use one-stop solution for neoantigen prediction from raw sequencing data. The main text has been revised to include this information (See section 2.6, page 6).

Round 3

Reviewer 2 Report

The authors properly addressed all of my comments.